# The Many Roles of Precision in Action

**DOI:** 10.3390/e26090790

**Published:** 2024-09-14

**Authors:** Jakub Limanowski, Rick A. Adams, James Kilner, Thomas Parr

**Affiliations:** 1Institute of Psychology, University of Greifswald, 17487 Greifswald, Germany; 2Institute of Cognitive Neuroscience, University College London, London WC1N 3AZ, UK; rick.adams@ucl.ac.uk (R.A.A.); j.kilner@ucl.ac.uk (J.K.); 3Centre for Medical Image Computing, University College London, London WC1N 6LJ, UK; 4Nuffield Department of Clinical Neurosciences, University of Oxford, Oxford OX1 4AL, UK; thomas.parr@ndcn.ox.ac.uk

**Keywords:** active inference, action, motor control, precision, predictive coding

## Abstract

Active inference describes (Bayes-optimal) behaviour as being motivated by the minimisation of surprise of one’s sensory observations, through the optimisation of a generative model (of the hidden causes of one’s sensory data) in the brain. One of active inference’s key appeals is its conceptualisation of precision as biasing neuronal communication and, thus, inference within generative models. The importance of precision in perceptual inference is evident—many studies have demonstrated the importance of ensuring precision estimates are correct for normal (healthy) sensation and perception. Here, we highlight the many roles precision plays in action, i.e., the key processes that rely on adequate estimates of precision, from decision making and action planning to the initiation and control of muscle movement itself. Thereby, we focus on the recent development of hierarchical, “mixed” models—generative models spanning multiple levels of discrete and continuous inference. These kinds of models open up new perspectives on the unified description of hierarchical computation, and its implementation, in action. Here, we highlight how these models reflect the many roles of precision in action—from planning to execution—and the associated pathologies if precision estimation goes wrong. We also discuss the potential biological implementation of the associated message passing, focusing on the role of neuromodulatory systems in mediating different kinds of precision.

## 1. Introduction

Active inference is a normative framework describing (Bayes-optimal) behaviour and cognition as motivated by the minimisation of surprise of one’s sensory observations [1]. This motivation is grounded in the Free Energy Principle [2], which posits that adaptive agents must occupy a limited region of states—separating internal from external states—to survive (statistically, this separation of self and non-self can be defined via a Markov blanket, cf. [3]). The agent can achieve this through minimising surprise (which, formally, can be approximated by a quantity called variational free energy) by aligning the predictions of its generative (predictive or “forward”) model and the sensory observations it receives. Thus, active inference can be formulated as a process theory, i.e., a mechanistic account of behaviour and the underlying brain functions, centred on the idea that the brain employs and optimises a model of the hidden causes of the sensory data it receives, approximating Bayesian inference [4]. As such, it falls into the category of Bayesian brain or “predictive processing” approaches, but it has a special place among those due to its scope and the detail of its formulation, including the mathematics of message passing and belief updating and their potential neurobiological implementation in cortical hierarchies [1,5,6].

Most notably, active inference extends beyond perceptual inference to explain action and behaviour in terms of inference within generative models; i.e., motor control is described as inference and model optimisation as well, where actions are generated to fulfil predictions, in parallel with the updating of predictions through perceptual inference. Thus, active inference is a framework with intriguing (and partly unique) assumptions about how the brain controls action and behaviour, which has attracted not only theoretical and experimental neuroscientists, but also psychologists and philosophers [7,8,9,10,11,12,13].

One of active inference’s key appeals is its conceptualisation of precision as defined within generative models, and its possible implementation in neuronal circuits. Precision, in short, is a form of confidence in the reliability of (neuronal) signals, which can bias or modulate neuronal communication and inference in the brain [4,14,15]. In other words, precision encodes “How much do I trust this information”. What exactly this means in each case—what it implies functionally—depends on the kind of information, the kind of computation, and the kind of brain region or network at play. Generative world models comprise a set of prior and conditional probability distributions whose inversion determines these computations. Precision is an attribute of all of these distributions and is associated with their inverse variance or negentropy. Thus, precision can mean the gain applied to corrective signals, i.e., prediction errors, at the level of sensation and movement—a notion that is important in engineering and motor control (e.g., [16,17]). Neurobiologically, this gain might manifest in population-level synaptic gains, which appear to be the loci of action of various neurochemical modulators [14,18,19]. At other levels, precision can mean a confidence estimate in which several world models are “correct” [20,21,22]. Both notions can be combined in hierarchical models for action—a particular promise of the active inference framework.

A key characteristic, independently of the level or kind of computation, is that precision must be estimated; i.e., the brain does not know from sensory signals alone how reliable they or their models are. Beliefs about precision are formed through inference by the generative model (sometimes called “second-order inference” to contrast it with inference about the causes of sensation per se; [23,24,25]). It is imperative for the brain to acquire these beliefs (precision estimates) correctly, otherwise many pathologies of action and perception emerge [26] (see below). Many papers have discussed the role of precision in perceptual inference, demonstrating the importance of ensuring precision estimates are right for normal (healthy) sensation and perception (e.g., [27,28,29,30,31]).

Here, we want to highlight the many roles precision plays in *action*, i.e., the many key processes in action that rely on adequate estimates of precision, from decision making and planning to the production of movement. We shall give a brief overview of empirical findings demonstrating the importance of precision estimation at various levels of action, from decision making and planning to the production of movement. Secondly, we shall discuss recent methodological developments in the generative models for active inference, which demonstrate the value of the precision concept in action. The concept of precision was, of course, not invented for or within the active inference (or predictive coding) framework. Still, in large part thanks to work by Karl Friston and his group, there are now very elegant ways to mathematically model precision in its various roles within generative models for active inference. Furthermore, these methods allow us to link the underlying equations to neuronal activity. We shall briefly introduce the key concepts of modelling precision (we give a non-mathematical overview, as the associated generative models have been introduced in great detail elsewhere; see, e.g., [1,15,32,33]). Of particular interest to us are recent developments of so-called “mixed” models—hierarchical generative models spanning multiple levels of discrete (categorical) and continuous active inference. These kinds of models open up new perspectives on the unified description of computations in action, spanning mechanisms from decision making and action planning to overt movement. We shall highlight how these models reflect the many roles of precision in action—from planning to execution—and the associated pathologies if precision estimation goes awry. Thereby, we shall also discuss the potential biological implementation of the associated message passing, focusing on the role of neuromodulatory systems in mediating precision.

## 2. Precision at Different Levels of Action

To highlight the importance of inference about precision and uncertainty in action, imagine the following example: You are a quarterback in possession of the football, and you want to make the optimal play to score your team some points. You must make several decision, such as: Which of the many plays you and your team have studied during training shall you initiate? Should you pass the ball now, or first run to a better spot? Then, of course, you must throw the pass in the best possible way, which means you must guide your muscle movements, correct your movement in response to sudden changes in the game such as a hostile player tackling you (i.e., you must know which sensations in your arm’s sensors you or another agent have generated, respectively), and many more factors. On top of this, you must constantly monitor the game, which you will probably use your vision and audition for. Sometimes, you can perhaps not very clearly see where your selected teammate is, but you hear them yelling instructions, so you focus your attention more on what you hear, perhaps neglecting vision. Figure 1 schematically illustrates this example.

This toy example is by no means thought to capture the entirety of action planning and motor control, but to simply illustrate that there are many different (e.g., sensory vs. motor vs. cognitive) processes in action that depend on adequate estimates of precision. Thereby, precision can mean different things: from an estimate of confidence in a selected action policy to a multiplicative gain applied to continuous sensory signals. Active inference offers a way to account for these potential differences by formalising action in hierarchical, “mixed” models. Roughly, these models comprise layers where inference operates in discrete time steps, or continually.

To understand why this is important, consider the fact that most decision making and action planning seem to be discrete or even categorical in nature, e.g., preparing a sequence of steps followed by throwing a pass, or choosing one among several alternative pre-studied plays. Conversely, sensation, attention, and movement generate continuous data: to play optimally, we need to know where our teammates are *now*, and where they will be next—in other words, we need to track their continuously changing states in the world. The same holds for guiding muscle movements. This is fundamentally different from categorical decisions such as selecting the optimal play among several alternatives, and from selecting the sequence of actions required to initiate the play, before actually committing to it. In the active inference framework, the underlying inference can be formalised slightly differently, operating in terms of discrete or continuous states (Figure 2). These different formalisations are more parsimonious for certain kinds of decisions or control; they also intuitively map onto differences in the associated mental or physical processes. We shall now give a brief, non-technical overview of the generative models used to capture this (for a mathematical treatment, see [32]).

First, it should be noted that, historically, active inference has been modelled as motor control in continuous time (using generalised coordinates of motion; i.e., speed, acceleration, jerk), based on an extension of predictive coding formulations of perceptual inference to include movement via spinal reflex arcs [34,35,36]. Now, as noted, the generative models used for active inference can be set up in an alternative way; i.e., they can also be formalised as operating in discrete states, modelling inference (belief updating) about processes that evolve in “steps” rather than continually. This notion of active inference follows a specific formulation of partially observable Markov decision processes, featuring a notably different computational architecture from, e.g., predictive coding or variational filtering. A particular distinction is the possibility to specify multiple alternative (counterfactual) goals and action sequences and to evaluate them based on their expected free energy, thus defining active inference as planning rather than motor control per se [32,37]. However, the discrete and continuous state space formulations are not mutually exclusive—quite the contrary: both conform to the free energy principle and can be described in terms of belief updating and model optimisation (for a comprehensive comparison, see [33]). The Bayesian message passing of both formulations can be plausibly associated with the neuronal populations in the cortical column, i.e., assigning different computations to different cell types in different cortical layers of the canonical microcircuit [5]. Simulations and computational models of empirical cortical responses have associated precision with the gain afforded to synaptic inputs onto pyramidal cell populations in supragranular (i.e., superficial) cortical layers [38,39,40]. In predictive-coding-inspired models, those cells are thought to communicate “bottom-up” information (i.e., prediction errors) to higher cortical levels [5]. Importantly, while neural activity represents different quantities in each model (posterior probabilities in discrete-time models vs. the statistics of probability densities in continuous-time models), precision can be linked to the gain afforded to synaptic inputs onto superficial pyramidal cells in both approaches [19]. However, each kind of model is best suited to a different set of computational problems that an active inference agent needs to solve and, as we shall see below, their combination is what is needed to adequately capture action in its entire richness.

## 3. Precision in Sensation and Movement (Continuous Inference)

Sensation and movement production are inherently intertwined, which is particularly emphasised in the active inference framework where sensation guides perceptual decision making for action, which, again, generates new sensory data. The world generates sensory signals continually, and to move within it, our brain should control our muscles continually as well. Consequently, the sensory and motor processes at the interface with the environment (the associated message passing) are best captured by continuous state space models, i.e., predictive models generating data (e.g., trajectories) in continuous time. A simple way to represent a trajectory in continuous time is in terms of generalised coordinates of motion (speed, acceleration, jerk, etc.), which has been the formulation of choice in these models [1]. In other words, these models capture the brain’s inference of how its sensations were caused by continuous states of the world [19]. As noted, this has been the classical way of modelling active inference in predictive coding schemes, i.e., through a dynamic minimisation of continuous prediction error signals [5,6,41].

Precision, in these formulations, has the function of scaling forward prediction errors at each level by their estimated reliability; i.e., sensory signals that are believed to be less noisy (more reliable) are afforded a greater weight [41]. This is thought to be implemented through changes in the synaptic gain of pyramidal cells in supragranular cortical layers communicating “bottom-up” signals [5,38,39]; interactions with inhibitory interneurons may also play a role [40]. Evidence for the functional role of the precision weighting of prediction errors has been obtained in electrophysiological and imaging studies and through mathematical simulations (e.g., [23,40,42,43]). This intuitively links precision at these levels of inference to the concept of sensory attention and attenuation [19,23,44]. Much empirical and theoretical work has focused on this role of precision as sensory gain [21,29,30,35,44,45,46,47,48]. In the following subsections, we focus on three important aspects of action, in which precision—formalised as continuous gain control—has been assigned a key role: multisensory body representation, movement production, and agency attribution or self–other distinction.

### 3.1. Sensory Attention and Attenuation in Multisensory Body Representation

Although not being a strictly “motor” process, multisensory integration is crucial for planning and guiding action, because it ensures a coherent and more precise estimate of the state of the body (effector) for guiding action. It is well established that a more accurate state estimate can be reached by combining information from multiple sensory modalities [49,50]. For instance, the quarterback in Figure 1 may combine visual and somatosensory (haptic and proprioceptive) information to grasp the football in a way that will allow, e.g., adding a specific spin to the throw (although this is probably something that benefits novices; expert players will likely not need to look at their arm or the ball anymore). Precision, in its role as sensory gain, can up- or down-weight sensory signals, thus augmenting or attenuating their impact on inference, i.e., on cue combination and the resulting multisensory estimate. In brief, the combined estimate will be biased towards the more precise (reliable) cue. That multisensory integration (or cue integration) can be biased by precision estimates has been demonstrated and reviewed extensively elsewhere [15,51,52,53]. This assumption is also at the core of Bayesian approaches to body ownership [8,9,54,55,56,57]. Here, a particularly intriguing example is the so-named rubber hand illusion [58], in which a participant experiences illusory ownership over a visible fake hand placed next to the real, unseen hand, and exhibits a recalibration of the perceived real hand position towards the fake hand. Bayesian models have captured the illusion based on a “dominance” of relatively more precise visual over (less precise) proprioceptive hand position estimates, which, together with a strong prior for having one hand only, leads to inference that the seen hand is “mine” [57,59,60]. Intriguingly, electrophysiological and brain imaging studies suggest that the brain could enhance the natural differences in precision between vision and proprioception to better resolve the multisensory conflict during the rubber hand illusion. In brief, these studies suggest that somatosensory information from the hand side subject to the illusion is selectively attenuated, while visual information in general seems to be up-weighted [61,62]. This would suggest a key role of top-down precision control in shaping the body representation, e.g., under unusual conflicts as in the rubber hand illusion [63].

Presumably, a similar sensory bias of the multisensory body representation by precision can also operate when we act. For instance, the relative weighting of visual vs. proprioceptive signals for the estimation of the body’s state can be biased top-down to augment visuomotor adaptation and learning. Several studies have suggested that, despite the importance of sensory prediction errors for motor learning, somatosensory attenuation during the early learning phase of visuomotor conflicts may enhance visuomotor adaptation—presumably by speeding up intersensory recalibration (e.g., [64]; see [63] for a review). Moreover, we could show that participants can deliberately change the relative weight assigned to visual vs. proprioceptive body position depending on their behavioural relevance during visuomotor conflict tasks, and that those changes can be seen in hemodynamic and oscillatory responses over the corresponding sensory cortices, much like one would expect gain control to operate along predictive coding formulations [65,66]. This further underwrites the importance of tuning precision estimates for flexible body representation; i.e., the context-dependent weighting and integration of (incongruent) seen and felt body position for action [57,67].

### 3.2. Sensory Attenuation for Movement Initiation

The second key role of precision as sensory gain pertains to movement production itself. This originates from active inference formulations based on an extension of predictive coding schemes with motor reflexes—specifically, from the key assumption of those formulations that movement is produced through the minimisation of proprioceptive prediction errors [34,35,36,47,68]. As noted above, in predictive processing approaches based on predictive coding, the model’s beliefs (probabilistic representations) capture statistical regularities in the environment and are optimised by accommodating prediction errors; this corresponds to perceptual inference. Active inference extends this idea to include movement; i.e., behaviour is explained in terms of inference on the causes of proprioceptive sensations [36]. Thus, descending signals from the primary motor cortex are conceptualised as proprioceptive predictions (i.e., about dynamic muscle or joint states [1,36]). Spinal reflex arcs, i.e., monosynaptic (or, in some cases, polysynaptic) loops of primary sensory afferents and motor neurons in the spinal cord, are then thought to minimise the error between predicted and actual proprioceptive states, whereby they cause muscle movements that approximate the predicted joint state [34]. Thus, the agent is now equipped with a complementary way to deal with prediction errors, as it can now act on the environment to directly reduce them. In other words, it can change the world (i.e., its sensory input) instead of changing its mind [1].

It should be noted that this “enaction” of proprioceptive predictions by the motor system is among the points of disagreement between active inference and optimal motor control [69,70]. In short, the assumption that descending signals from the primary motor cortex are proprioceptive “predictions” rather than motor “commands” eliminates the need for a cortical “inverse” model, which, in optimal control formulations, computes the motor commands required to reach the action goal [69]. Instead, the “pure forward model” approach of active inference [1,70] relegates the inverse model to spinal reflex arcs, which reduce the prediction error resulting from the mismatch of predicted and actual proprioceptive data [36,68]. Despite these differences, however, both accounts are largely compatible, as they converge on the notion that actions (and movements) are predicted and controlled largely by internal generative (“forward”) models in the brain [70,71]. Likewise, both accounts emphasise the importance of sensory attenuation, albeit with different underlying mechanisms.

Here, active inference entertains one unique assumption, namely, that sensory attenuation is required to move. This results directly from the aforementioned conceptualisation of behaviour in terms of inference on the causes of proprioceptive sensations, where the spinal reflex arc minimises proprioceptive prediction errors. Note that descending proprioceptive predictions, as assumed by active inference to drive muscle movements (see above), are per se counterfactual; i.e., I am not really moving, and not actually receiving those proprioceptive signals that the descending “motor” signals predict. Along the classical formulation of (perceptual) inference, the resulting prediction error should be accommodated by updating the model beliefs; i.e., changing my predictions. This, however, would result in immobility, as movement would no longer be predicted. The solution to this problem is thought to lie in sensory attenuation, i.e., in selectively suppressing the precision afforded to proprioceptive signals. As a result, the descending proprioceptive predictions are now thought to be dominant. The organism does not update them, but chooses another way to directly minimise the prediction error: it changes the proprioceptive data itself. This means nothing else than that the organism *moves*, thus generating the predicted proprioceptive trajectory. In short, in active inference, sensory attenuation is a requirement for movement initiation. This is to the extent that a failure to correctly initiate or maintain movement can be modelled as a failure of adequate sensory attenuation. There is considerable experimental evidence demonstrating sensory attenuation during movements and at the time of movement initiation consistent with active inference. For movements, somatosensory attenuation has typically been studied using the electrical stimulation of the median nerve. This produces a somatosensory evoked potential (SSEP) recordable at multiple levels of the somatosensory pathway to provide a measure of the magnitude of the afferent volley. Cortical EEG recordings have shown that there is a suppression of the primary and secondary complexes of the SSEP during active and passive movement [72]. The attenuation of SSEPs has also been shown during motor preparation before the EMG onset of active movement [73,74]. Furthermore, patients with Parkinson’s Disease who have bradykinesia (a deficit in movement initiation and maintenance of movement) show significantly reduced or no attenuation of the primary SSEP component when off dopaminergic medication, but this is restored when administered such medication [75,76].

In short, in active inference, sensory attenuation is the complement of sensory attention. To produce movement, the agent has to attend away from proprioceptive data by attenuating their gain, decreasing the confidence in them. Note that this does not mean that the agent is uncertain about motor control or movement itself, but, momentarily, about its sensations [47]. This may be mediated by low-frequency neuronal synchronisation [77,78].

Interestingly, this account offers an explanation for certain phenomena observed under visuo-proprioceptive conflicts, as in the rubber hand illusion (see above), i.e., the generation of apparently involuntary unseen hand movements towards the location of a displaced fake hand (observed empirically and replicated in simulations [79]). In brief, in the active inference approach, the seen hand location could bias proprioceptive predictions of the felt, unseen hand position; the involuntary hand movements can be seen as an “enaction” of those proprioceptive predictions [80]. Something similar may also happen in motor contagion and visuomotor interference, and in related confusions of “agency”, as described in the following subsection.

### 3.3. Sensory Attenuation of Reafference and Its Relation to Agency

It is well documented that sensory reafference (self-generated movement feedback) is relatively suppressed [81,82]. Within the active inference framework, this can be explained by the fact that the precision of the corresponding sensory signals is attenuated to allow movement (see above). This implies a key role of precision control (sensory attenuation) in self–other distinction [43,63,83]. This becomes particularly evident when considering disturbances of agency attribution. For the present purposes, we roughly define “agency” as the experience or feeling of being in control of “one’s” actions per a self-attribution of movement and its sensory consequences. Many accounts of pathological behaviour have focused on aberrant predictive mechanisms, casting atypicalities in sensation and perception in, e.g., psychosis or in the autism spectrum as prediction failure [27,28,29,84,85,86,87]. In short, along active inference, the distinction between self and other, in part, relies on the fact that “self-generated” sensations (i.e., those predicted by the motor system) are attenuated in order to allow movement (see above), whereas externally generated sensations are not [43,44]. Thus, a relatively attenuated proprioceptive sensation can be a cue that I am performing a voluntary act, while enhanced proprioceptive signals would, conversely, signal that someone or something else caused my movement [44]; potentially, this can be extended to other types of reafference, e.g., vision [88]). This renders precision estimation a key feature of agency attribution and self–other distinction in social settings. Correspondingly, simulation studies have shown that a number of characteristic phenomena in schizophrenia/psychosis can be explained by an imbalance in precision between higher hierarchical areas (where it is too low) and lower—e.g., primary sensory—areas (where it is too high). These include impaired smooth-pursuit eye movements, reduced evoked responses to unexpected stimuli in sensory oddball tasks, reduced sensitivity to some visual illusions, and, in particular, a loss of attenuation of self-generated sensations (reviewed in [89]). The latter has been demonstrated in the force-matching task [90], although note that self-produced sensations could be magnified both by reduced attenuation (increased precision) and/or by impaired prediction (increased prediction errors), and few studies test both possibilities. Evidence has been found of both problems in schizotypy [91]. Heightened sensations, including from somatic senses, can often result in delusions about foreign objects or external influences on the body in schizophrenia. Interestingly, patients with functional neurological disorders who suffer from symptoms such as paralysis or dystonia without evidence of neurological damage also show reduced sensory attenuation in the force-matching task [46], implying that loss of agency (for self-generated motor symptoms) may have a similar mechanism across these disorders [30].

A final interesting showcase of the importance of precision for self-agency and self–other distinction can be found in the phenomena of motor contagion or visuomotor interference. As mentioned, the relative precision assigned to visual and proprioceptive information determines its impact on the brain’s multisensory estimate of hand posture and position (see above). Achieving the balance between these two senses is key for bodily identification and self–other distinction, as it determines whether a seen movement is simply observed (other body) or whether it is executed (own body; [43,68,83]). This explains why observing the incongruent movements of conspecifics biases our own movement execution [66,92,93,94]. Similarly, new-borns first show a pronounced imitation of observed movements, which could be explained by an automatic activation of a body representation that does not yet distinguish between self (i.e., movements associated with visual and proprioceptive consequences) and other (i.e., movements associated with only visual consequences [12,56]). In certain psychiatric and neurological conditions, such as catatonic schizophrenia, Tourette’s syndrome, or after prefrontal lesions, a similar “echopraxia” can be observed. This can be thought of as a loss of control. In active inference formulations, specifically, it implies a loss of control over one’s expectations of sensory precision, resulting in aberrant attention and attenuation, i.e., an inadequate weighting of sensory evidence that causes an automatic update of a body model and, in some cases, may lead to explicit misattributions of agency and pathological self-experience [48,95,96]. This renders the execution of control over sensory precision (the context-dependent selective emphasis or suppression of sensory evidence, i.e., sensory attention and attenuation) a key mechanism of bodily self-identification and self–other distinction [63]. Yet, at the same time, it enables us to be empathetic and to understand our conspecifics’ action intentions [43].

## 4. Precision in Action Planning (Discrete Inference and Mixed Models)

Above, we summarised how body movements are produced according to continuous (active) inference. This can be seen as “driving action selection in the present to change currently available sensory data” [1] (p. 9). In contrast to this, an alternative formulation of active inference accommodates planning, i.e., inference to select the optimal course of action in the future [32,37].

The prominent way of modelling this is as inference in discrete time via hidden Markov models or partially observable Markov decision processes (see Figure 2A). Roughly, the corresponding models comprise policies (which represent sequences of actions), hidden states, and observable sensory data generated by the latter. As states evolve over time according to transition matrices, each action plan (policy) can be evaluated in terms of how likely it is to generate preferred observations. This anticipated fulfilment of preferences is evaluated as part of the policy’s expected free energy (for details, see, e.g., [32,97]). The other aspect of expected free energy deals with a prior belief that policy selection will lead to the resolution of any uncertainty, offering an exploratory complement to the “exploitative” preference fulfilment. In these formulations, precision can be associated with beliefs about uncertainty in the mapping of outcomes onto hidden states, in the state transitions, or about action policies [19]. As we shall see below, these different forms of uncertainty can be elegantly linked to different processes of action planning.

The preference for a different formulation arises from the fact that, in contrast to continuous-time joint movements and the sensations driving them, decision making and action planning are categorical in nature; therefore, their underlying processes are very intuitively modelled in discrete time [98,99]. In fact, one can even argue that very soon after leaving the level of sensory receptors, one can speak of discretised representations in the neuronal hierarchy, such as classical receptive fields [1].

Practically all actions require planning *and* moving, i.e., inference in discrete and continuous state spaces. Even our simple example in Figure 1 requires the selection of an optimal action policy and the enaction of the movement, i.e., the predicted proprioceptive states. How can this be accommodated, and how can these levels interact? This has recently been addressed within the active inference framework through the development of hierarchical, “mixed” generative models [32]. These models encompass more than one level of belief updating, encompassing both discrete and continuous-state spaces (see Figure 2B for an example). This is needed to capture inference underlying action because action entails several processes ranging from discrete (categorical) decisions to continuous sensation and movement (see Figure 1). Thus, in such mixed models, discrete inference will occupy the higher levels of the hierarchy, while continuous inference will underwrite sensation and movement. This means that the lowest level of these models will necessarily be continuous, because it is the agent’s interface with the world in continuous time [1]. Yet, these continuous levels must link to categorical states, as discrete action planning is informed by continuous sensory data, and (discretely) selected actions generate sequences of (continuous) movements.

In discrete as well as continuous formulations (Figure 2A,B), precision is thought to be implemented via the weighting of synaptic inputs. However, the locus of this synaptic weighting varies depending upon the probability distribution with which that precision is associated. In other words, while the role of precision as synaptic gain control is universal, the association of precision with different distributions in the (brain’s) generative model means that “precision” will mean something slightly different at each level of the model hierarchy. For those precisions associated with likelihoods (i.e., conditional distributions linking hierarchical levels), we might anticipate an effect on superficial pyramidal cells that mediate the ascending connections from one cortical region to another. However, for precisions over (for example) policies we might pursue, the relevant synapses might be expected in the basal ganglia, where monoamines such as dopamine can modulate the actions selected. Fast changes in synaptic efficacy such as these (as opposed to slower processes like learning) can plausibly be linked to the action of neuromodulatory systems in the brain (see below).

In line with this, we have recently applied such a hierarchical mixed model to capture the computational architecture underlying sequential, goal-directed pointing movements, encompassing multiple discrete levels modelling target selection and action planning, and a continuous level generating joint (arm) movements [100]. By applying simulated lesions to different precision parameters, we could reproduce behavioural changes resembling those associated with either anatomical lesions or neuromodulatory pathologies, mapping the computational architecture onto the known anatomy of movement (Figure 3A). We briefly summarise the main findings in the following section.

The idea behind this work was to simulate a simple task in which movement planning and execution are necessary, but can be disambiguated from one another. Three targets were placed in fixed locations in a 3D space, with one highlighted as the target to aim for. Periodically, the highlighted target would change. The highest level of the generative model dealt with transitions between the highlighted target states, with each time step predicting a short sequence of discrete positions for one’s hand at the level below. Finally, the discrete hand positions at each time point were used to predict short trajectories in continuous time, using the generalised coordinates of motion apparatus alluded to above. During the inversion of this model, the expected free energy was used to select alternative hand positions, which, via the continuous model at the lowest level, led to proprioceptive predictions that were fulfilled through (reflexive) actions.

Several forms of precision were important here. At the continuous level, this included two key types of precision. The first was the inverse variance of the anticipated sensory data (“sensory precision”). When this was increased—as if the descending corticospinal tracts that attenuate precision during movement (see above) were interrupted—we found relatively normal movement, but greatly exaggerated tendon reflexes elicited by introducing an unexpected proprioceptive stimulus (see Figure 3B). This is typical of so-called ‘upper motor neuron’ pathologies, which range from stroke [101] to motor neuron disease [102]. The second precision at the continuous level determined the anticipated smoothness of sensory fluctuations (i.e., their autocorrelation over time). The augmentation of this precision resulted in hypermetric and ataxic-like behaviour of the sort associated with cerebellar syndromes [103,104].

Neither of the above precision parameters affected the ability to decide upon the correct (highlighted) target, just the execution of the movement to reach that target. In contrast, manipulations to decrement the precision of policy selection led to an almost akinetic picture, in which no target could be selected with any degree of confidence and the simulated arm remained somewhere in the middle. This might reflect the difficulty in motor initiation seen in Parkinsonian patients with dopamine depletion [105]. Finally, we found that reducing the precision that linked between the highest hierarchical level—dealing with changes in the highlighted target—and the level dealing with discrete sequences of hand positions resulted in a form of perseveration, in which every change in target position was associated with a very delayed (but ultimately successful) change in the motor plan to reach the new target. We might think of this form of precision attenuation as the effect of a frontal disconnection syndrome [106], with disconnection being the most extreme form of loss of synaptic gain [107,108]. Figure 3B shows the respective simulated pointing trajectories.

In sum, the hierarchical mixed models briefly described above offer a great tool to investigate the neurocomputational basis of action. Specifically, they provide the possibility to simulate lesions to certain brain systems, e.g., those tentatively associated with the implementation of precision, as described in the following section.

## 5. How Are Different Kinds of Precision Implemented in the Brain?

One of the appealing characteristics of active inference is that it can be a process theory, i.e., a mechanistic description of how Bayesian message passing can be mapped onto neurobiological computations [1]. Inter alia, this allows the generation of empirically testable predictions based on the assumed generative model that our brains are using, which may ultimately inform the construction of artificial agents [16,17,109]. Thus, the inferential (Bayesian) message passing in active inference can be mapped onto the canonical cortical microcircuit [5,6]. This can be achieved for continuous-time formulations, as in predictive coding, and discrete-time formulations such as partially observable Markov decision processes, albeit with some subtle differences [1,33].

In both formulations, precision estimation can be cast as gain control, reflecting the confidence in probability distributions, e.g., the weight afforded to prediction error signals or the confidence in a policy or likelihood mapping. The associated fast changes in synaptic efficacy (as opposed to those related to slower learning processes) can, in principle, be linked to neuromodulation [1,18,19,110,111]. In particular, the focus on neuromodulation allows a very intuitive distinction between the different kinds of precision in action [14,19].

Among the numerous neurotransmitters of the central nervous system, some have received particular attention in this formulation; with respect to precision and uncertainty, those are acetylcholine, dopamine, and noradrenaline (or norepinephrine [14,112,113,114]. The cholinergic, dopaminergic, and noradrenergic systems are often referred to as “neuromodulatory” systems, in contrast to neurotransmitters that are classically assigned a primarily excitatory or inhibitory role, such as Glutamate or GABA (although these can also influence attentional processes by acting on, e.g., cholinergic projections; cf. [114]). This is due to several characteristics of the respective neurons and their synaptic effects; for instance, in contrast to, e.g., Glutamate or GABA, neurons producing acetylcholine, dopamine, or noradrenaline are predominantly located in a few nuclei in the brainstem, from where they have more or less widespread connections to large parts of the brain [14,111]. This means that a single neuron can influence processes in many (e.g., cortical) neurons simultaneously—a role that lends itself to a global modulatory mechanism such as precision [19]. Figure 4 shows a schematic of the main cholinergic, dopaminergic, and noradrenergic pathways of the brain.

Broadly, the *cholinergic* system can be seen as a top-down control mechanism, ultimately originating from the prefrontal cortex through its control over cholinergic neurons, e.g., in the basal forebrain, which modulates processing in sensory cortices [112]. One of its likely roles is in the control of (selective) attention [38,115]. Thus, it has been established that cholinergic neuromodulation can be modelled as enhancing sensory or “expected” precision [14,38]. In discrete-time models, this would correspond to the likelihood precision, i.e., how confident one is that hidden states will cause certain observable outcomes [19]. In our example (Figure 1), cholinergic modulation would be at play at the lower sensorimotor levels such as sensory attention and attenuation.

The *dopaminergic* system has classically been linked to movement production and action selection, in large part because its main pathways target predominantly motor structures in the basal ganglia [35,100]. Active inference models have also linked dopaminergic activity to the precision of cues that afford action [35], which, in discrete-time formulations, corresponds to the precision of action policies—i.e., the confidence in action plans [19,116]. In our example, dopaminergic modulation would influence the selection of a specific play and movement.

Finally, the *noradrenergic* system seems to encode the certainty in (precision of) model predictions where noradrenergic responses of the brainstem locus coeruleus signal low certainty. Put differently, a substantially increased noradrenergic signalling from the locus coeruleus would signal “global model failure” [22] in light of unexpected errors [14]; i.e., it signals that the current model is gravely wrong and needs updating in light of incoming sensory evidence. This becomes relevant in learning and model updating, particularly when learning depends on the estimated volatility of the environment [117]. One way to conceptualise this is that noradrenaline deals with confidence in how predictably the world changes (i.e., it signals precision in transition probabilities)—a view that is evidenced by modelling of pupillometric data during the online manipulation of volatility [118]. Evidence for this key role of noradrenergic precision estimates comes from the study of autism spectrum conditions: computational models of empirical behaviour suggest that the overly “precise” perception—and the resulting over-reactivity to environmental changes and corresponding difficulties in adaptive learning—in the autism spectrum may result from hyperactive noradrenergic signalling [86,119]. These results fit very nicely with predictive coding accounts of brain function, specifically, with the idea of noradrenaline encoding precision of model predictions [87,120].

Joint investigations of all three of the above neuromodulatory systems are still rare, but one computational study using pharmacological manipulations of all three neuromodulators overall confirmed the association of acetylcholine with uncertainty about cue-outcome associations; dopamine with action selection; and noradrenaline with learning from unexpected changes [113]. The role of serotonin in attention and precision is less clear [114,121], but it may be linked to preferences or the precision of interoceptive likelihoods [1]. In sum, there are still many outstanding questions about the neuromodulatory mechanisms of precision to be answered by future work.

Of course, neuromodulation is only one possible way to mediate precision estimates; for instance, they could also be computed and communicated through the induction of oscillatory coherence, i.e., synchronisation between cortical areas [6,47,122]. For instance, low-frequency oscillations in the “alpha” range have classically been assigned a functional role in mediating top-down (selective) sensory attention [78,123]. In many studies, attention to stimuli in various sensory modalities has been linked to a suppression of alpha power in the respective sensory cortical areas (besides augmenting gamma power). Similar attentional suppression has been observed over sensory cortices in the neighbouring “beta” frequency range [115,124,125]. However, beta oscillations originating from the motor system seem to have a different functional role. Thus, Palmer et al. [47,77] proposed that the desynchronisation in the beta range typically observed during movement reflects an increased uncertainty in the current proprioceptive signals. In other words, this interpretation assigns sensorimotor beta oscillations a role in sensory attenuation, in line with the assumption that an attenuation of proprioceptive signals is required to move (see above). This interpretation of motor beta oscillations as mediating sensory attenuation potentially complements the role of sensory alpha/beta oscillations in sensory attention.

Importantly, the association of different kinds of precision with different neuromodulators and the synchronisation of different frequency bands is not mutually exclusive. On the contrary, there is good evidence to assume a related mechanism where, for example, cholinergic mechanisms could mediate the attention-related synchronisation of neuronal populations via its inhibitory effect on interneurons [112,115]. More work is needed to establish such a relationship, and to draw a clear precision-related functional distinction between different neuromodulators and neuronal synchronisation in different frequency bands, but the above studies are valuable first steps.

## 6. Conclusions and Outlook

To conclude, the precision concept has proven very valuable on many levels of describing brain function and behaviour, from cognitive control and decision making to sensory attenuation and motor reflexes. Of course, the reviewed contributions have not answered all questions, and they have opened up new ones. A major line of open questions surely pertains to the implementation of active inference and precision in the brain. Some of the key assumptions of the framework are disputed, and there is a need for properly designed empirical studies that contrast them with other accounts of brain function and motor control [6,33,126]. For instance, gain modulation may be associated with more (and/or different) computations than changing precision [127,128]. An exciting empirical research direction is the link between neuromodulators, neuronal synchronisation, and the different kinds of precision discussed above. One specific question in this line of research pertains to the timing of precision in active sampling, i.e., involving the coordination of action and effector movements operating on different time scales [97,129] and a potential “entrainment” of sensory attention by behavioural rhythms [130,131]. We did not discuss inference underlying the balance between explorative and exploitative behaviour, but it is likewise an important decision for action where adequate precision estimates (e.g., as confidence in control) are essential and determine behavioural flexibility, switching between tasks, and balancing the impact of habits vs. goals in action planning [99,132,133].

In the field of schizophrenia/psychosis research, a crucial open question is the mechanism behind the various precision changes that one can infer from behaviour. For example, higher hierarchical (e.g., prefrontal or hippocampal) loss of precision could be due to reduced synaptic gain from NMDA receptor dysfunction [134], or reduced neuromodulation—e.g., of the prefrontal cortex by dopamine 1 receptors [135] or muscarinic receptors [136]—but reduced attenuation in sensory areas may be due to disinhibition (loss of interneuron function) or poorly understood actions of dopamine in the sensory striatum [137]. Conversely, other studies have found that striatal dopamine activity seems to strengthen the precision of prior beliefs about sensations, not sensory input [138,139], a possible mechanism for hallucinations and a likely contributory factor to delusions [140].

A very important field to which the concept of precision in generative models can contribute on many levels are cyber–physical interactions, i.e., interactions involving the control over robotic or virtual bodies (avatars). Robotics itself is a specific field of application, where Bayesian (active) inference can be implemented to control action [16,17,109]. Equipping artificial agents with separate estimates in, e.g., action policies and likelihood mappings may prove useful for generating truly human-like behaviour. This includes social–cognitive processes like action understanding, agency attribution, and joint action, which all rely on learning the appropriate balance between sensory attention and attenuation [43,83,93]. More generally, cyber–physical interaction means learning control over alternative bodies and, often, in different “realities”, as adopting control over a virtual or remote-controlled robotic body will provide the user with different sensory information and different degrees of freedom for movement. Thus, the adoption of a virtual or robotic body can be likened to a switch to a different reality: a different world and a different body representation [63]. This poses many complex problems to the embodied self [12,141,142], including how to adjust one’s precision estimates on several levels of inference for optimal (flexible) behaviour.

We have already emphasised the importance of tuning precision estimates for aiding visuomotor adaptation, i.e., the integration of novel seen and felt bodily information for action (see above and [63,94,115,124]). The same likely factors into adaptation to a different sensorimotor mapping in cyber–physical interactions. Furthermore, precision can act as sensory attention and attenuation to enhance immersion: an attenuation of distracting signals from the “physical reality” could help attend to those of the desired (virtual) reality; this can mean simple things such as a person playing virtual reality games ignoring the weight or limited field of view of the virtual reality headset, or visuomotor lags introduced by systematic delays in the hardware. But it is unknown what the limits and consequences of this learning are. For instance, in which of these cases is the model (body representation) adjusted, when is a new model learned, and how efficiently can we switch between models? One interesting question is whether there is some sort of sensory attenuation when switching between physical and virtual body representations (or vice versa), similar to what has been discussed for movement initiation above. There are further potential roles of precision in cyber–physical interaction, related to perceptual inference. For instance, estimates of *interoceptive* precision can mediate a sense of “presence”, such as being *in* the virtual reality [13,143]. This could perhaps link to a higher-level role of precision as model certainty (see above), undermining the sense that the current world- and self-model is “right” [8,20]. In short, there are many possible roles of precision when switching between physical and virtual body representations and, correspondingly, many theoretical and empirical questions that can and should be addressed from within the active inference framework.

## Figures and Tables

**Figure 1 entropy-26-00790-f001:**
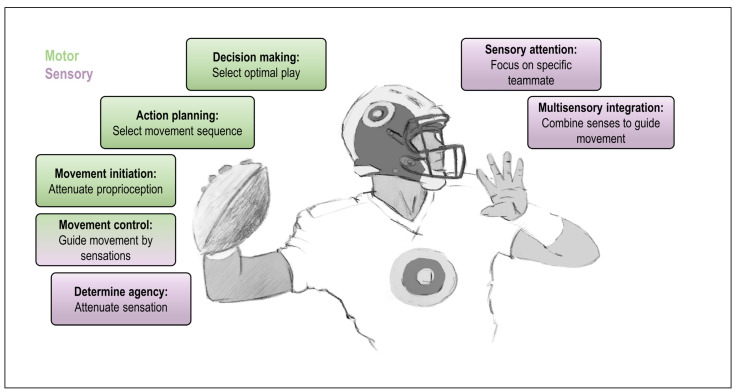
Some processes of an action where precision estimates are essential: A toy example shows a quarterback passing the football to a specific teammate indicating several important components of (active) inference that rely on adequate estimates of precision. The underlying computations cover processes ranging from decision making (e.g., Which play do I select? Where should I run to in order to be able to pass optimally?) to overt movement (contractions of the appropriate arm muscles throughout the throwing movement) and many more that are not shown, including motivational factors and habits, action understanding, joint action, and communication. Precision plays a key role in all of these processes, but a somewhat different one depending on the exact nature of inference. For instance, at “higher” cognitive levels, the player must decide which of several pre-studied plays he initiates. Here, one can describe precision as the confidence in the selected (optimal) sequence of actions. At “lower”, e.g., sensorimotor levels, precision can be described as a multiplicative gain on sensory signals. This can mean implementing sensory attention when selectively focusing on one particular teammate, and a similar bias in determining the weights of sensory cues during multisensory integration. Multisensory integration is essential to guide action, e.g., integrating visual and proprioceptive body position information to guide movement, or integrating seen and heard information about a teammates’ location. Not least, this notion of precision is key to how muscle movement is produced and controlled along the active inference framework: sensory attenuation is a prerequisite for the enaction of motor predictions and a potential clue for determining agency and self–other distinction. Note that some of the illustrated processes can be cast as based on discrete or even categorical inference (such as deciding on one among several plays), whereas others require inference in continuous time to track continuous trajectories of sensory data coming from the world (such as guiding a movement or attending to data from a particular sensory channel). Active inference offers a framework to model action through the combination of discrete and continuous state space models, thus capturing the interplay between the illustrated cognitive vs. sensorimotor processes, and the different roles of precision therein.

**Figure 2 entropy-26-00790-f002:**
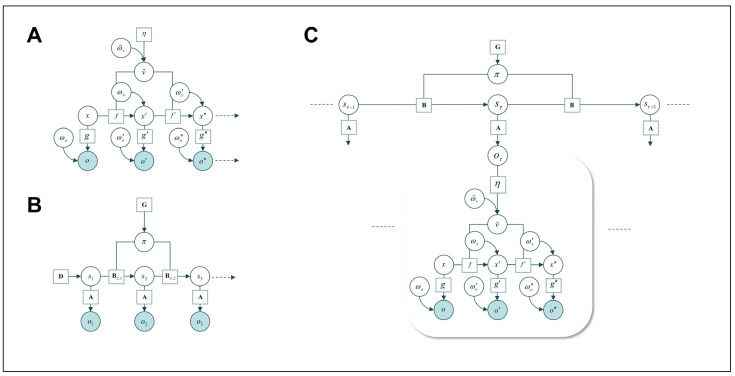
Continuous, discrete, and mixed models for (active) inference. (**A**): Inference in continuous time via a continuous state space model in terms of generalised coordinates of motion. This kind of model generates data (i.e., trajectories) in continuous time, using generalised coordinates of motion (speed, acceleration, jerk, etc.) to represent the trajectory. The details of this model are explained in [32]. The key point here is that *a continuous trajectory* of sensory observations *o* (*o*′, *o*″,…, corresponding to speed, acceleration, etc.) is modelled as caused by a hidden state *x* and its derivatives (*x*′, *x*″,…, where the interactions between the temporal derivatives are determined by an equation of motion *f* prescribed by a hidden cause *v*) through a nonlinear mapping *g*, plus random fluctuations *ω*. This elegantly captures the fact that the world generates sensory inputs continually, and, furthermore, that we act upon the world through continuous muscle movements. For this reason, these formulations are typically used to model sensation and movement, for instance, based on prediction error minimisation in predictive coding schemes. (**B**): Inference in a discrete state space formulation. The key difference to the model shown in (**A**) is that we are seeing a sequence of three distinct hidden states *s*_1_–*s*_3_, which each generate corresponding an observable outcome *o*_1_–*o*_3_ through a matrix (**A** specifying the likelihood mapping). The states are linked by transition matrices **B**, which, in turn, depend on the current policy (sequence of actions encoded by *π*; **G** represents the probability distribution over policies based on expected free energy; **D** represents the initial state; see [32]). In contrast to the trajectory generated by the model in (**A**), this model generates data in discrete steps. These formulations lend themselves to model discrete or even categorical inference of the sort that, presumably, guides decision making or action planning. (**C**): “Mixed” model of action comprising a discrete state space level sitting “on top” of, and linked to, a continuous state space level, each displayed as a Bayesian network. The upper discrete level generates “chunks” of data in discrete time (the Bayesian network represents conditional dependencies) and, thus, models categorical decisions or discrete action plans; the lower continuous level generates data in continuous time (the Bayesian network represents generalised coordinates of motion). The link between the levels happens as the outcomes of the discrete model determine a hidden cause that prescribes the generalised motion of continuous hidden states, generating continuous sensation. Here, the upper level could select an optimal action sequence (such as a particular throwing movement), which allows the generation of muscle movements through proprioceptive predictions via the lower level (thus, actually throwing the football). Precision estimates play an important, but different, role in several computations at both levels of this model (see Figure 1 and main text). Adapted from [32], Figures 1, 5 and 8 under the CC-BY 4.0 licence.

**Figure 3 entropy-26-00790-f003:**
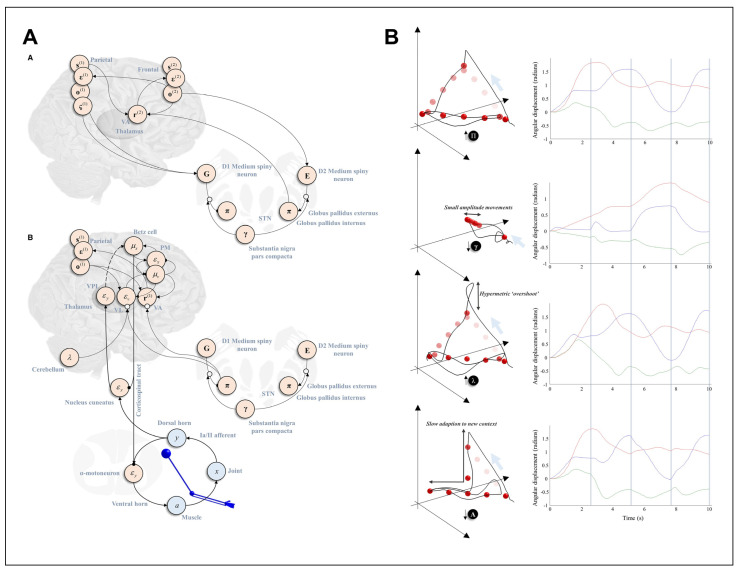
Using hierarchical active inference to simulate action and its pathologies. (**A**): Mapping inferential message passing onto the known anatomy of movement. Here, to simulate pointing movements to three visual targets, we used a hierarchical mixed model with two linked discrete levels, inferring pointing sequences and intermediate attracting points for movement, respectively; the lower level linked to a continuous level, as in Figure 2. The top schematic (**small A**) shows the mapping of two discrete levels of the mixed model, concerned with target and action selection, onto frontoparietal cortices and structures of the basal ganglia. The bottom schematic (**small B**) shows the relationship between the lower discrete level and the continuous level of the model, which ultimately issues proprioceptive predictions that are enacted by movement through spinal reflexes in continuous time. For details, see [100]. (**B**): This architecture was used to simulate pointing movements to three visual targets under different synthetic lesions. The black lines in the left plots show the trajectory of the simulated arm; the red spheres represent the sequence of attracting points selected by the (lower) discrete model that determine short trajectories for the continuous model (reminiscent of the concept of motor “chunking”). The right plots show the corresponding changes in shoulder rotation and flexion, and elbow flexion. From top to bottom: Overestimation of sensory precision did not impair movement, but exaggerated tendon reflexes (not shown). Reducing the precision of the beliefs about action policy selection produced “akinetic”, small-amplitude movements. The overestimation of the anticipated smoothness of sensory fluctuations over time produced hypermetric overshoots at the end of each movement. Finally, reducing the precision associated with linking the discrete model levels concerned with target and action policy selection, respectively, produced an apparent confusion whenever the target position changed. Reprinted from (Figures 5 and 6 in [100]), under the CC-BY 4.0 licence.

**Figure 4 entropy-26-00790-f004:**
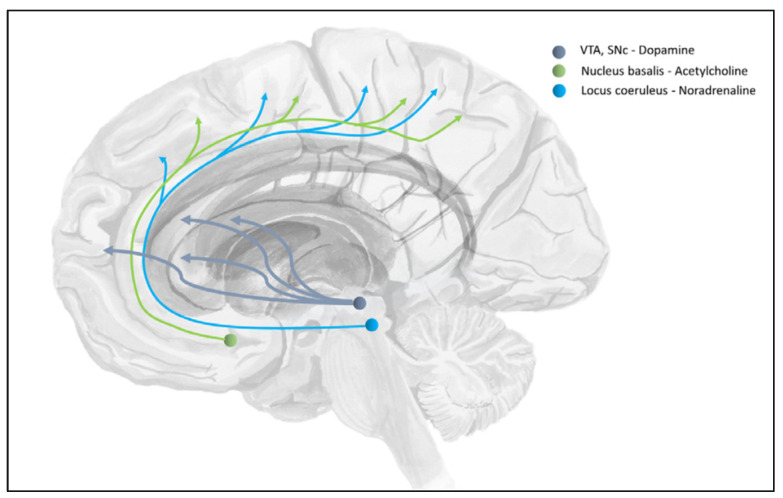
Neuromodulatory systems associated with precision in action. The cholinergic, dopaminergic, and noradrenergic pathways have been linked to mediating different kinds of precision or uncertainty. Within the active inference framework, these neuromodulators can be linked to the precision afforded to sensory signals, action policies and control, and model predictions (about the dynamics of changes in the environment), respectively. Figure reprinted from (Figure 4 in [110]), under the CC-BY 4.0 licence. VTA = ventral tegmental area, SNc = substantia nigra pars compacta.

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
