# Peer review of "The Many Roles of Precision in Action"

_entropy, 2024, doi:10.3390/e26090790_

Round 1

Reviewer 1 Report

Comments and Suggestions for Authors

Dear Authors,

Your review entitled "The many roles of precision in action" provide an interesting update on how precision weights hierarchical processing signaling during action planning and execution. Focusing on recent generative models that reflect this precision weighting in action, you describe how simulating lesions on different forms of precision signaling allowed you to investigate the computational architecture of action. You nicely review the different pathways involved in precision weighting of synaptic inputs e.g. the different forms of precision weighting by distinct neuromodulators and precision weighting of forward hierarchical signaling that appears later in the text to be exerted via top-down alpha or beta synchronization, depending on the sensory system involved. You then conclude with work to be done and future implementation for cybernetics application. One aspect that I felt missing is a brief description of the current anatomical knowledge about where in the local microcircuit top-down precision signals meet and interact with bottom-up forward signaling.

Sincerely

Author Response

Dear Authors,

Your review entitled "The many roles of precision in action" provide an interesting update on how precision weights hierarchical processing signaling during action planning and execution. Focusing on recent generative models that reflect this precision weighting in action, you describe how simulating lesions on different forms of precision signaling allowed you to investigate the computational architecture of action. You nicely review the different pathways involved in precision weighting of synaptic inputs e.g. the different forms of precision weighting by distinct neuromodulators and precision weighting of forward hierarchical signaling that appears later in the text to be exerted via top-down alpha or beta synchronization, depending on the sensory system involved. You then conclude with work to be done and future implementation for cybernetics application. One aspect that I felt missing is a brief description of the current anatomical knowledge about where in the local microcircuit top-down precision signals meet and interact with bottom-up forward signaling.

Sincerely

Response: Thank you very much for your kind words, and for the important suggestion. We have extended our descriptions of this in lines 130ff. as follows:

“The Bayesian message passing of both formulations can be plausibly associated with the different neuronal populations in the cortical column; i.e., assigning different computations to different cell types in different cortical layers of the canonical microcircuit (Bastos et al., 2012). Simulations and computational models of empirical cortical responses have associated precision with the gain afforded to synaptic inputs onto pyramidal cell populations in supragranular (i.e., superficial) cortical layers (e.g., Moran et al., 2013; Pinotsis et al., 2017; cf. Auksztulewicz & Friston, 2015). In predictive coding inspired models, those cells are thought to communicate ‘bottom-up’ information (i.e., prediction errors) to higher cortical levels (Bastos et al., 2012). Importantly, while neural activity represents different quantities in each model (posterior probabilities in discrete-time models vs the statistics of probability densities in continuous-time models), precision can be linked to the gain afforded to synaptic inputs onto superficial pyramidal cells in both approaches (Parr & Friston, 2017).”

And in lines 152ff.:

“Precision, in these formulations, has the function of scaling forward prediction errors at each level by their estimated reliability; i.e., sensory signals that are believed to be less noisy (more reliable) are afforded a greater weight (Friston & Kiebel, 2009). This is thought to be implemented through changes in the synaptic gain of pyramidal cells in supragranular cortical layers communicating ‘bottom-up’ signals (Bastos et al., 2012; Moran et al., 2013; Pinotsis et al., 2017; interactions with inhibitory interneurons may also play a role, cf. Auksztulewicz & Friston, 2015). Evidence for the functional role of precision weighting of prediction errors has been obtained in electrophysiological and imaging studies and through mathematical simulations (e.g., Kok et al., 2012; Auksztulewicz et al., 2015; Feldman and Friston 2010; Friston and Frith 2015).”

Please also see our previous description in lines 376ff. for further clarification:

“In discrete as well as continuous formulations (Fig. 2A-B), precision is thought to be implemented via the weighting of synaptic inputs. But the locus of this synaptic weighting varies depending upon the probability distribution with which that precision is associated.  In other words: While the role of precision as synaptic gain control is universal, the association of precision with different distributions in the (brain’s) generative model means that ‘precision’ will mean something slightly different at each level of the model hierarchy. For those precisions associated with likelihoods (i.e., conditional distributions linking hierarchical levels) we might anticipate an effect on superficial pyramidal cells that mediate the ascending connections from one cortical region to another. However, for precisions over (for example) policies we might pursue, the relevant synapses might be expected in the basal ganglia, where monoamines such as dopamine can modulate the actions selected. Fast changes in synaptic efficacy such as these (as opposed to slower processes like learning) can plausibly be linked to the action of neuromodulatory systems in the brain (see below).”

Added references:

  • Pinotsis, D. A., Geerts, J. P., Pinto, L., FitzGerald, T. H., Litvak, V., Auksztulewicz, R., & Friston, K. J. (2017). Linking canonical microcircuits and neuronal activity: Dynamic causal modelling of laminar recordings. Neuroimage146, 355-366.

Hopefully, these changes are what you had in mind. Thank you again for your comments!

Reviewer 2 Report

Comments and Suggestions for Authors

This is article has been well-thought out, and although it is hard for me to be critical of it, I do have some brief comments. Since from an early stage, active inference has been emphasized, it was surprising that the Free Energy Principle (FEP) and the relationship to the Markov blanket (MB) concept, seemed to have been muted somewhat. Perhaps include mention of the FEP as a least action principle, whereby systems with sufficient interaction with their environments can be considered as enclosed by an MB. The dictum here is no doubt familiar to the authors, noting that minimization of Variational Free Energy, results in minimizing prediction error which seems relevant to the theme of the article. 

Regarding sensory attenuation, there is fairly wide literature on pathologies of prediction, precision, and attunement as these apply to the clinical study of autism spectrum conditions. It would be fitting to mention this, say in L281 et seq., or L433 et seq.  Suggested references are: 

Pellicano, E. and Burr, D.  When the world becomes ‘too real’: a Bayesian explanation of autistic perception. Trends. Cogn. Sci.  16 (2012), 504–510.   Lawson, R.P., Rees, G. and Friston, K. J.  An aberrant precision account of autism. Front. Hum. Neurosci. 8 (2014), 302.   (the latter pinpointing pyramidal neurons)   Van de Cruys, S., Evers, K., Van der Hallen, R., Van Eylen, L., Boets, B., de-Wit, L. and Wagemans, J. Precise minds in uncertain worlds: predictive coding in autism. Psychol. Rev. 121 (2014), 649–675.  

Author Response

This is article has been well-thought out, and although it is hard for me to be critical of it, I do have some brief comments.

Response: Many thanks for your very positive evaluation of our manuscript, and for your helpful comments. We have copy-pasted our corresponding changes to the text below. Hopefully this is what you had in mind.

Since from an early stage, active inference has been emphasized, it was surprising that the Free Energy Principle (FEP) and the relationship to the Markov blanket (MB) concept, seemed to have been muted somewhat. Perhaps include mention of the FEP as a least action principle, whereby systems with sufficient interaction with their environments can be considered as enclosed by an MB. The dictum here is no doubt familiar to the authors, noting that minimization of Variational Free Energy, results in minimizing prediction error which seems relevant to the theme of the article. 

Response: Thank you for this suggestion. We have added this motivation at the very beginning of our manuscript (lines 18ff.):

“Active inference is a normative framework describing (Bayes optimal) behaviour and cognition as motivated by the minimisation of surprise of one’s sensory observations (Parr et al., 2022). This motivation is grounded in the Free Energy Principle (Friston, 2010), which posits that adaptive agents must occupy a limited region of states—separating internal from external states—to survive (statistically, this separation of self and non-self can be defined via a Markov blanket, cf. Kirchhoff et al., 2018). The agent can achieve this through minimising surprise (which, formally, can be approximated by a quantity called variational free energy) by aligning the predictions of its generative (predictive or “forward”) model and the sensory observations it receives. Thus, active inference can be formulated as a process theory; i.e., a mechanistic account of behaviour and the underlying brain functions: centred on the idea that the brain employs and optimises a model of the hidden causes of the sensory data it receives, approximating Bayesian inference (Friston, 2005). As such, it falls into the category of Bayesian brain or “predictive processing” approaches—but it has a special place among those due to its scope and the detail of its formulation, including the mathematics of message passing and belief updating and their potential neurobiological implementation in cortical hierarchies (Parr et al., 2022; cf. Bastos et al., 2012; Shipp, 2016).”

Added references:

  • Friston, K. (2010). The free-energy principle: a unified brain theory?. Nature reviews neuroscience11(2), 127-138.
  • Kirchhoff, M., Parr, T., Palacios, E., Friston, K., & Kiverstein, J. (2018). The Markov blankets of life: autonomy, active inference and the free energy principle. Journal of The royal society interface15(138), 20170792.

Regarding sensory attenuation, there is fairly wide literature on pathologies of prediction, precision, and attunement as these apply to the clinical study of autism spectrum conditions. It would be fitting to mention this, say in L281 et seq., or L433 et seq.  Suggested references are: 

Pellicano, E. and Burr, D.  When the world becomes ‘too real’: a Bayesian explanation of autistic perception. Trends. Cogn. Sci.  16 (2012), 504–510.   Lawson, R.P., Rees, G. and Friston, K. J.  An aberrant precision account of autism. Front. Hum. Neurosci. 8 (2014), 302.   (the latter pinpointing pyramidal neurons)   Van de Cruys, S., Evers, K., Van der Hallen, R., Van Eylen, L., Boets, B., de-Wit, L. and Wagemans, J. Precise minds in uncertain worlds: predictive coding in autism. Psychol. Rev. 121 (2014), 649–675.  

Response: Many thanks for your very positive evaluation of our manuscript, and for your helpful comments.

Lines 290ff.:

“Many accounts of pathological behaviour have focused on aberrant predictive mechanisms; casting atypicalities in sensation and perception in e.g. psychosis or in the autism spectrum as prediction failure (Fletcher & Frith, 2009; Bansal et al., 2018; Corlett et al., 2009; Adams et al., 2012; Pellicano & Burr, 2012; Lawson et al., 2014; Van de Cruys et al., 2014).

Lines 433ff. i.e. 481:

“Finally, the noradrenergic system seems to encode the certainty in (precision of) model predictions; where noradrenergic responses of the brainstem locus coeruleus signal low certainty. Put differently, a substantially increased noradrenergic signalling from the locus coeruleus would signal “global model failure” (Jordan, 2023) in light of unexpected errors (Yu & Dayan, 2005); i.e., it signals that the current model is gravely wrong and needs updating in light of incoming sensory evidence. This becomes relevant in learning and model updating, particularly when learning depends on the estimated volatility of the environment (Sales et al., 2019). One way to conceptualise this is that noradrenaline deals with confidence in how predictably the world changes (i.e., it signals precision in transition probabilities)—a view that is evidenced by modelling of pupillometric data during online manipulation of volatility (Vincent, Parr et al. 2019). Evidence for this key role of noradrenergic precision estimates comes from the study of autism spectrum conditions: Computational models of empirical behaviour suggest that the overly ‘precise’ perception—and the resulting over-reactivity to environmental changes and corresponding difficulties in adaptive learning—in the autism spectrum may result from hyperactive noradrenergic signalling (Lawson et al., 2014, 2021). These results fit very nicely with predictive coding accounts of brain function, specifically, with the idea of noradrenaline encoding precision of model predictions (Van de Cruys et al., 2014; Yon, 2021).

Added references:

  • Lawson, R. P., Rees, G., & Friston, K. J. (2014). An aberrant precision account of autism. Frontiers in human neuroscience8, 302.
  • Pellicano, E., & Burr, D. (2012). When the world becomes ‘too real’: a Bayesian explanation of autistic perception. Trends in cognitive sciences16(10), 504-510.
  • Van de Cruys, S., Evers, K., Van der Hallen, R., Van Eylen, L., Boets, B., De-Wit, L., & Wagemans, J. (2014). Precise minds in uncertain worlds: predictive coding in autism. Psychological review121(4), 649.
  • Yon, D. (2021). Prediction and learning: Understanding uncertainty. Current Biology31(1), R23-R25.

We hope these changes are what you had in mind – thank you again for your helpful suggestions.